# The Activity of KIF14, Mieap, and EZR in a New Type of the Invasive Component, Torpedo-Like Structures, Predetermines the Metastatic Potential of Breast Cancer

**DOI:** 10.3390/cancers12071909

**Published:** 2020-07-15

**Authors:** Tatiana S. Gerashchenko, Sofia Y. Zolotaryova, Artem M. Kiselev, Liubov A. Tashireva, Nikita M. Novikov, Nadezhda V. Krakhmal, Nadezhda V. Cherdyntseva, Marina V. Zavyalova, Vladimir M. Perelmuter, Evgeny V. Denisov

**Affiliations:** 1Laboratory of Cancer Progression Biology, Cancer Research Institute, Tomsk National Research Medical Center, Russian Academy of Sciences, 634009 Tomsk, Russia; t_gerashchenko@oncology.tomsk.ru (T.S.G.); zolotarevasyu@oncology.tomsk.ru (S.Y.Z.); kiselev_am@almazovcentre.ru (A.M.K.); novikovnm@oncology.tomsk.ru (N.M.N.); 2Institute of Cytology, Russian Academy of Sciences, 194064 Saint Petersburg, Russia; 3Department of General and Molecular Pathology, Cancer Research Institute, Tomsk National Research Medical Center, Russian Academy of Sciences, 634009 Tomsk, Russia; tashireva@oncology.tomsk.ru (L.A.T.); pk_ssmu@ssmu.ru (M.V.Z.); pvm@ngs.ru (V.M.P.); 4Department of Pathological Anatomy, Siberian State Medical University, 634050 Tomsk, Russia; kaf.pat.anatom@ssmu.ru; 5Laboratory of Molecular Oncology and Immunology, Cancer Research Institute, Tomsk National Research Medical Center, Russian Academy of Sciences, 634009 Tomsk, Russia; nvch@tnimc.ru

**Keywords:** breast cancer, metastasis, invasion, heterogeneity, morphology, torpedo-like structures

## Abstract

Intratumor morphological heterogeneity reflects patterns of invasive growth and is an indicator of the metastatic potential of breast cancer. In this study, we used this heterogeneity to identify molecules associated with breast cancer invasion and metastasis. The gene expression microarray data were used to identify genes differentially expressed between solid, trabecular, and other morphological arrangements of tumor cells. Immunohistochemistry was applied to evaluate the association of the selected proteins with metastasis. RNA-sequencing was performed to analyze the molecular makeup of metastatic tumor cells. High frequency of metastases and decreased metastasis-free survival were detected in patients either with positive expression of KIF14 or Mieap or negative expression of EZR at the tips of the torpedo-like structures in breast cancers. KIF14- and Mieap-positive and EZR-negative cells were mainly detected in the torpedo-like structures of the same breast tumors; however, their transcriptomic features differed. KIF14-positive cells showed a significant upregulation of genes involved in ether lipid metabolism. Mieap-positive cells were enriched in genes involved in mitophagy. EZR-negative cells displayed upregulated genes associated with phagocytosis and the chemokine-mediated signaling pathway. In conclusion, the positive expression of KIF14 and Mieap and negative expression of EZR at the tips of the torpedo-like structures are associated with breast cancer metastasis.

## 1. Introduction

Invasion is the first step towards cancer metastasis [1,2]. During the invasion, cancer cells penetrate surrounding tissue and then intravasate into blood vessels, where they spread via blood flow to distant organs, extravasate and form the micrometastases, which finally outgrow into macroscopic metastases [1].

According to the classical concept, there are two modes of cancer cell invasion: single-cell (individual) and collective invasion [3]. The type of cell movement is defined by cellular polarity, actin cytoskeleton organization, and stability of cell–cell junctions [4,5,6]. Single-cell invasion is strongly associated with a process of epithelial–mesenchymal transition (EMT), during which cancer cells lose epithelial characteristics and obtain mesenchymal traits, namely increased motility and invasiveness [7,8]. In collective invasion, cancer cells are undergoing a partial EMT when mesenchymal traits are acquired but epithelial features and intercellular contacts are not lost [6,9]. Collectively invading cells can be part of large tumor masses or be formed into groups with the different architectural organization [3,10].

Although the mechanisms of cell migration and invasion have been described quite well, there are currently no effective markers for the identification of invading cancer cells and, therefore, for assessment of the invasive potential of tumors [11]. These markers could be used to identify patients at the high risk of metastasis and to prescribe therapy aimed at interrupting the metastatic process. In addition, these markers might represent targets for future therapeutics that block invasion and metastasis [11].

Invasive carcinoma of no special type (invasive ductal carcinoma), the most common form of breast cancer, demonstrates significant intratumor morphological heterogeneity. Breast cancer cells may be single or arranged in either small groups (two to five cells), collectively named discrete groups of tumor cells, or multicellular structures: tubular, alveolar, solid, and trabecular structures (Figure 1). These morphological structures have been found to represent transcriptionally distinct tumor cell populations with varying degrees of EMT and invasiveness and to be associated with breast cancer prognosis. Tubular and alveolar structures are transcriptionally similar and demonstrate similar expression of epithelial and mesenchymal markers. Solid structures show an increase in mesenchymal traits but retain epithelial features. Trabecular structures, small groups of tumor cells, and single tumor cells display a pronounced mesenchymal phenotype and a dramatic decrease in epithelial traits [12,13]. Breast cancers with trabecular structures or single tumor cells show increased metastatic potential [14,15]. Based on these results, we assumed that tubular and alveolar structures show decreased invasive potential, whereas solid and trabecular structures, as well as single tumor cells, are highly invasive [12]. It is important to note that solid structures are morphologically heterogeneous and represented by arrangements with either small bud-like or large torpedo-like sprouts. The last ones probably can detach from solid structures and transform into torpedo-like structures [11] (Figure 1).

In this study, we aimed to identify the molecules associated with breast cancer invasion and metastasis. The data of gene expression profiling of solid and trabecular structures, Gene Set Enrichment Analysis, and the Human Protein Atlas were used to select genes associated with cell migration and with heterogeneous expression at the protein level in breast cancer. Immunohistochemistry (IHC) was used to evaluate the association of the selected proteins with metastasis. The modified immunostaining procedure, fluorescence-guided laser microdissection, and RNA-sequencing were used to investigate the molecular makeup of tumor cells associated with metastasis.

## 2. Results

### 2.1. Molecules Potentially Associated with Breast Cancer Invasion

Using the gene expression data of breast cancer morphological structures (GEO, GSE80754), we selected genes potentially associated with cancer invasion based on the following criteria:

1. Differentially expressed genes (DEGs) in solid and trabecular structures as compared to normal breast epithelium (*p* < 0.1) (Appendix A) and their association with cell migration according to Gene Set Enrichment Analysis [16] (Appendix A).

2. Genes expressed in solid and trabecular structures but not in tubular and alveolar structures (*p* < 0.05) and their association with cell migration according to literature data (Appendix A).

3. Expression of proteins encoded by the selected genes at the periphery and the tips of the solid and/or trabecular structures according to the Human Protein Atlas [17] (Appendix A).

In total, four proteins were selected: HCLS1 associated protein X-1 (HAX1), kinesin family member 14 (KIF14), mitochondria-eating protein/spermatogenesis associated 18 (Mieap/SPATA18), and ezrin (EZR) (Table 1).

### 2.2. Positive Expression of KIF14 and Mieap and Negative Expression of EZR Are Associated with Breast Cancer Metastasis

Based on the IHC data, we assessed the association of the expression of HAX1, KIF14, Mieap, and EZR with breast cancer metastasis. No significant association was found for HAX1. In contrast, KIF14, Mieap, and EZR expression were significantly correlated to metastasis. Distant metastases were more often detected in patients with expression of KIF14 and Mieap at the tips of the torpedo-like structures in breast tumors as compared to KIF14- and Mieap-negative cases (66.7% versus 14.3%, *p* = 0.003 and 70.0% versus 12.5%, *p* = 0.001, respectively; Table 2). Distant metastases were also more frequently detected in patients with negative expression of EZR at the tips of the torpedo-like structures than in EZR-positive cases (73.7% versus 13.3%, *p* = 0.001; Table 2). Metastasis-free survival was decreased in patients with positive expression of KIF14 (HR 6.63, 95% CI: 1.50–29.29, *p* = 0.013) and Mieap (HR 8.09, 95% CI: 1.83–35.75, *p* = 0.006) and negative expression of EZR (HR 7.65, 95% CI: 1.72–33.93, *p* = 0.007) at the tips of the torpedo-like structures (Figure 2).

It must be noted that torpedo-like structures were detected in 54.7% (46/84) of patients and just their presence was not associated with breast cancer metastasis. In particular, distant metastases were observed in 43.8% (20/46) patients with torpedo-like structures and 50.0% (19/38) patients without these structures (*p* = 0.55). Distant metastasis was not also related to the positive expression of KIF14 and Mieap and negative expression of EZR in other morphological arrangements of tumor cells: tubular, alveolar, solid and trabecular structures and small groups of tumor cells (data not shown). In addition, no significant association was found between expression of KIF14, Mieap, and EZR and molecular subtype of breast cancer, tumor size, grade, lymph node metastases, and recurrence.

### 2.3. Patterns of Expression of KIF14, Mieap, and EZR in Breast Cancer

The representative images of IHC staining for KIF14, Mieap, and EZR proteins are provided in Appendix A. In total, positive expression of KIF14 and Mieap and negative expression of EZR were observed in 58.4 (45/77), 45.4% (35/77), and 59.2% (45/76) of breast cancers, respectively. Positive expression of KIF14 and Mieap and negative expression of EZR were more often observed in trabecular (74.3–91.7%) and tubular structures (62.5–88.2%) as compared to other structures and single tumor cells (39.1–60.0%; *p* < 0.05; Appendix A).

### 2.4. Transcriptomic Profile of KIF14-Positive Tumor Cells Located in Torpedo-Like Structures

We found 260 DEGs (*p* < 0.05) including 105 upregulated and 155 downregulated genes between KIF14-positive and KIF14-negative tumor cells located in torpedo-like structures (Appendix A). Among the most upregulated genes were *NHS* (NHS Actin Remodeling Regulator), *CNOT2-DT* (long non-coding RNA PRANCR), *THBS4* (thrombospondin 4), and others. The *SLITRK6* gene was the most downregulated gene in KIF14-positive cells (Figure 3A, Appendix A). According to Kaplan–Meier plotter [18], low expression of *NHS*, *THBS4,* and *SLITRK6* is associated with poor relapse-free survival (RFS) in breast cancer patients (Appendix A). The upregulated genes were predominantly related to ether lipid metabolism, whereas the downregulated genes were mostly involved in AGE-RAGE and T cell receptor signaling pathways (Figure 3B). The GO annotation revealed that upregulated genes are prevalently associated with collagen fibril organization, whereas downregulated genes were related to regulation of cellular protein localization and metabolic processes (Figure 3B). The complete list of Kyoto Encyclopedia of Genes and Genomes (KEGG) and Gene Ontology (GO) terms enriched in KIF14-positive cells is given in Appendix A.

### 2.5. Transcriptomic Profile of Mieap-Positive Tumor Cells Located in Torpedo-Like Structures

A total of 226 DEGs (106 upregulated and 120 downregulated at *p* < 0.05) were identified in Mieap-positive tumor cells as compared to Mieap-negative tumor cells located in torpedo-like structures (Appendix A). The most significantly upregulated genes included *CCDC18* (Coiled-Coil Domain Containing 18) and *SAMD9* (Sterile Alpha Motif Domain Containing 9). Among the most downregulated genes were *FOSB* (FosB Proto-Oncogene), *GPR153* (G Protein-Coupled Receptor 153), *MYH4* (Myosin Heavy Chain 4), and others (Figure 4A). According to Kaplan–Meier plotter [18], low expression of *FOSB*, *GPR153,* and *MYH4* is associated with poor RFS in breast cancer patients (Appendix A). The upregulated genes were enriched for the sulfur relay system, mitophagy, and synthesis of heterocycles (e.g., molybdopterin cofactor (Figure 4B)). The downregulated genes were associated with the Hedgehog signaling pathway, morphogenesis of an epithelial tube, mammary gland development, and other KEGG and GO terms (Figure 4B). The complete list of KEGG and GO terms enriched in Mieap-positive cells is given in Appendix A.

### 2.6. Transcriptomic Profile of EZR-Negative Tumor Cells Located in Torpedo-Like Structures

There were 162 DEGs (98 upregulated and 64 downregulated at *p* < 0.05) between EZR-negative and EZR-positive tumor cells located in torpedo-like structures (Appendix A). The most upregulated genes included *CD109*, *ID4* (Inhibitor Of Differentiation 4), *ST8SIA1* (ST8 Alpha-N-Acetyl-Neuraminide Alpha-2,8-Sialyltransferase 1), *NR4A1* (Nuclear Receptor Subfamily 4 Group A Member 1), and others. The most significantly downregulated gene was *SYCP3* encoding synaptonemal complex protein 3 (Figure 5A). According to Kaplan–Meier plotter [18], low expression of *CD109, ID4, NR4A1*, and *SYCP3* is associated with poor RFS in breast cancer patients (Appendix A). The upregulated genes were involved in the regulation of phagocytosis, chemokine-mediated signaling pathway, and synthesis and secretion of several hormones, whereas the downregulated genes were mainly associated with homologous recombination and beta-alanine metabolism (Figure 5B). The complete list of KEGG and GO terms enriched in EZR-negative cells is given in Appendix A.

### 2.7. KIF14- and Mieap-Positive and EZR-Negative Cells are Co-Localized in Torpedo-Like Structures

It turned out that positive expression of KIF14 and Mieap and negative expression of EZR were mainly observed in torpedo-like structures of the same breast tumors. Simultaneous expression of all three proteins was observed in 57.9% (11/19) of the patients with metastases. The co-expression of any two proteins was detected in 15.8% (3/19) of the cases (Appendix A). Multiplex IHC staining demonstrated that KIF14 and Mieap positive expression and EZR negative expression were both observed in the same cells and were specific for different cells within the tips of the torpedo-like structures (Figure 6A). Nevertheless, RNA-sequencing showed no overlapping DEGs (*p* < 0.05) between KIF14-positive, Mieap-positive, and EZR-negative cells (Figure 6B). Low overlapping was found when all genes up- and downregulated in KIF14-positive, Mieap-positive, and EZR-negative cells were analyzed (Figure 6C).

### 2.8. Expression of Classic Markers of Invasion in KIF14-Positive, Mieap-Positive, and EZR-Negative Cells

We checked whether Ki-67, urokinase-type plasminogen activator (uPA/PLAU), urokinase plasminogen activator surface receptor (uPAR/PLAUR), and matrix metalloproteinases (MMP2, MMP9, and MMP13) are expressed in KIF14-positive, Mieap-positive, and EZR-negative cells located in torpedo-like structures. No significant up- or downregulation of genes encoding the above-mentioned proteins were found. The exception was the downregulation of the *MMP13* gene in KIF14-positive cells as compared to KIF14-negative cells (Appendix A). Nevertheless, multiplex IHC staining showed no differences in MMP13 expression between tumor cells composing the torpedo-like structure (Appendix A).

## 3. Discussion

Metastasis is a hallmark of malignant tumors and is responsible for 90% of cancer-related deaths [2]. Understanding the mechanisms of cancer cell movement, the discovery of markers for assessment of invasive and metastatic potential, and the development of the therapeutics for prevention of cancer dissemination are some of the main challenges in oncology. In this study, we used the intratumor morphological heterogeneity in breast cancer, mainly solid and trabecular arrangements of tumor cells possessing mesenchymal and invasive traits [12], as a model for the identification of molecules associated with breast cancer invasion and metastasis. Our results indicate that positive expression of KIF14 and Mieap and negative expression of EZR at the tips of the torpedo-like structures are significantly associated with breast cancer metastasis. KIF14- and Mieap-positive and EZR-negative cells were mainly detected in the torpedo-like structures of the same breast tumors; however, their transcriptomic features differed. This probably indicates the presence of three functionally-distinct types of tumor cells at the tips of the torpedo-like structures. It is important to note that distant metastasis was not associated either with just the presence of torpedo-like structures or KIF14, Mieap, and EZR expression in other morphological structures. Thus, the activity of KIF14, Mieap, and EZR at the tips of the torpedo-like structures predetermines the metastatic potential of breast cancer and KIF14- and Mieap-positive and EZR-negative tumor cells are most likely potential metastasis-initiating cells.

Torpedo-like structures have been suggested by us recently and are one of the morphological manifestations of the invasive component in breast cancer [11]. These structures represent the formations of an irregularly elongated, mainly triangular shape, consisting of two to three parallel rows of cells, and have a wide base and a pointed tip of up to three cells. In this study, torpedo-like structures were detected in 54.7% of the breast cancers. The origin of torpedo-like structures and their place in the evolution of intratumor morphological heterogeneity are a subject for further research. Most likely, these tumor cell arrangements detach from solid structures with torpedo-like sprouts and evolve to trabecular structures [11]. Thus, together with solid and trabecular structures, torpedo-like formations of tumor cells can reflect patterns of collective cell invasion. In this regard, KIF14- and Mieap-positive cells, as well as EZR-negative cells, located at the tips of the torpedo-like structures, can possess migratory and invasive phenotype.

Kinesin-14 (KIF14) is known to play a role in intracellular transport [19] and is implicated in cytokinesis [20]. In breast cancer, an increased level of KIF14 is related to high tumor grade and poor overall survival [21]. The knockdown of KIF14 inhibits migration and invasion of breast tumor cells in vitro [22]. In our study, KIF14-positive cells located at the tips of the torpedo-like structures show a significant enrichment of upregulated genes associated with ether lipid metabolism. Previously, it was found that increased ether lipid level in tumor cells is correlated with the aggressiveness of cancer [23]. KIF14-positive cells also demonstrate the significant upregulation of the *NHS* gene encoding a novel regulator of actin remodeling and cell morphology [24] and long non-coding RNA *CNOT2-DT* (PRANCR) regulating cell cycle progression and clonogenicity [25]. In addition, KIF14-positive cells show upregulation of the *THBS4* (thrombospondin 4) gene that is known to possess proangiogenic and proinflammatory activity in breast cancer [26,27]. Moreover, the *NHS* and *THBS4* gene expression are associated with RFS in breast cancer patients according to Kaplan–Meier plotter [18].

Mitochondria-eating protein (Mieap) controls mitochondrial quality by repairing or eliminating unhealthy mitochondria [28,29]. The accumulation of lysosomal proteins in the mitochondria in a Mieap-dependent manner results in mitochondrial autophagy or mitophagy [30]. Mitochondrial quality control plays a pivotal role in cancer migration and invasion [31]. Disruption of mitophagy processes leads to the accumulation of reactive oxygen species in the tumor microenvironment, thereby modulating cancer cell growth, migration, invasion, and metastasis [29,31,32]. As expected, Mieap-positive cells located at the tips of the torpedo-like structures are enriched in genes involved in mitophagy (*BNIP3*, *RRAS2*, and *PGAM5*). These tumor cells also show the significant downregulation of the *FOSB* gene whose depletion was recently found to promote proliferation and growth of triple-negative breast cancer cells by inactivating the p53 pathway [33]. In addition, the *FOSB* gene is associated with RFS in breast cancer patients according to Kaplan–Meier plotter [18].

EZR is a member of the ezrin-radixin-moesin (ERM) family of proteins, which function as cross-linkers between the plasma membrane and the actin cytoskeleton. Many studies showed that ezrin is involved in the regulation of focal adhesion and invadopodia dynamics [34], and its overexpression is associated with metastasis of various cancers including breast cancer [35,36,37]. Surprisingly, we found that EZR negative expression at the tips of the torpedo-like structures is associated with breast cancer metastasis. EZR-negative tumor cells are enriched in the phagocytosis and the chemokine-mediated signaling pathway. One of the phagocytosis-associated genes, *TLR2* encodes Toll-like receptor 2, which is involved in the release of inflammatory cytokines and facilitation of cancer invasion and metastasis [38,39]. The chemokine-mediated signaling pathway is represented by *SLIT2* and *SLIT3* genes that play a suppressive role in tumor metastasis [40] and are often epigenetically silenced in a wide variety of cancers including breast cancer [41,42,43]. EZR-negative tumor cells also show the significant upregulation of *CD109*, *ID4*, *ST8SIA1*, and *NR4A1* genes. In breast cancer, CD109, ID4, and ST8SIA1 are involved in the maintenance of stem cell phenotype [44,45,46], whereas NR4A1 promotes TGF-β-induced EMT and invasion [47]. Besides, these genes are associated with RFS of breast cancer patients according to Kaplan–Meier plotter [18].

The mechanisms underlying the differential activity of KIF14, Mieap, and EZR in torpedo-like structures are unknown and further research is required. Probably, non-genetic factors may account for changes in the expression of these proteins, because, according to TCGA, *KIF14*, *EZR*, and *SPATA18* genes are very rarely mutated in breast cancer (less than 1%) [48]. It is also unclear if KIF14- and Mieap-positive and EZR-negative cells interact with each other in torpedo-like structures and whether their cooperation is needed for breast cancer invasion and metastasis. According to previous studies, cellular cooperation is integrally important for effective cancer invasion [49,50,51,52]. Nevertheless, despite these issues, the activity of KIF14, Mieap, and EZR in torpedo-like structures presents an interest for further understanding mechanisms of invasion–metastasis cascade, and can be a potential marker for predicting the risk of breast cancer metastatic spread. In this regard, our results emphasize that several cancer markers together rather than in their singularity could provide a valuable tool for the “metastatic stratification” of breast cancer patients, as previously reported by Massague and colleagues [53,54,55]. In other words, metastasis always involves the cooperation of several genes.

The study has several limitations, and the findings should be interpreted with caution. The study group is small and the results should be validated in independent and large cohorts. In addition, research in vitro is required to confirm whether positive expression of KIF14 and Mieap, as well as negative expression of EZR, are critical for breast cancer migration and invasion. In this regard, it would be important to develop the 3D in vitro model that would mimic the intratumor morphological heterogeneity of breast cancer including torpedo-like structures.

## 4. Materials and Methods

### 4.1. Patients

Eighty-four patients with IC NST (T_1-4_N_0-3_M_0_), between 31 and 78 years of age (mean age: 50.7 ± 9.88), were treated in the Cancer Research Institute, Tomsk NRMC (Tomsk, Russia) from 1991 to 2015. The clinicopathological parameters of breast cancer patients are provided in Appendix A.

The formalin-fixed, paraffin-embedded (FFPE) samples of breast tumor tissue were used for immunohistochemistry analysis. The frozen tumor and normal tissue specimens were used for RNA sequencing.

The procedures followed in this study were in accordance with the Helsinki Declaration (1964, amended in 1975 and 1983). All patients signed informed consent for voluntary participation. The study was approved by the review board of the Cancer Research Institute, Tomsk NRMC on 19 March 2015 (the approval number is 3).

### 4.2. Gene Expression Microarrays

The gene expression microarray data (GEO, GSE80754) of tubular, alveolar, solid, and trabecular structures, as well as discrete groups of tumor cells, were evaluated using the R software (R Development Core Team, 2008) and the limma package from BioConductor [56]. Log mean spot signals were taken for further analysis. Expression levels were normalized to normal breast epithelium. The transcripts were ranked for differential expression using a moderated t-statistic as implemented in the limma package.

### 4.3. Immunohistochemistry

IHC was applied to assess HAX1, KIF14, Mieap, and EZR expression in breast tumors (n = 84). Seven-micrometer-thick sections of FFPE tumor samples were deparaffinized, rehydrated, and stained as previously described [57]. The following antibodies were used: Rabbit anti-HAX1 (NBP1-54800, 1:50, Novusbio, Centennial, CO, USA), rabbit anti-KIF14 (HPA038061, 1:500, Sigma, St. Louis, MO, USA), rabbit anti-Mieap/SPATA18 (HPA036854, 1:100, Sigma, St. Louis, MO, USA), and rabbit anti-EZR (HPA021616, 1:1000, Sigma, St. Louis, MO, USA). The stained sections were assessed for expression of HAX1, KIF14, Mieap, and EZR in different morphological structures of breast tumors. In particular, the assessment included the presence of structures (tubular, alveolar, solid, and trabecular) with positive and negative cells at their periphery (up to 2 layers), the presence of tumor cells with positive and negative expression at the tips (up to 3 rows) of the torpedo-like sprouts connected with solid structures and the torpedo-like structures (not connected with solid structures), positive and negative expression in single tumor cells, etc. The complete study protocol is given in Appendix A. In total, 10 to 24 fields of view were analyzed per sample. Expression was counted as positive if it was observed in at least 10 structures of the same type.

### 4.4. Multiplex Immunohistochemistry

Multiplex IHC was used to analyze the co-localization of KIF14, Mieap, and EZR proteins in torpedo-like structures. Multiplex IHC was performed with a Bond RXm system (Leica, Hamburg, Germany) with antibodies against KIF14 (HPA038061, 1:500, Sigma, St. Louis, MO, USA; detected by Opal 520), Mieap/SPATA18 (HPA036854, 1:100, Sigma, St. Louis, MO, USA; Opal 620), EZR (HPA021616, 1:1000, Sigma, St. Louis, MO, USA; Opal 690), and MMP13 (MA5-14238, 1:25, Thermo Fisher Scientific, Waltham, MA, USA). Protein blocking was performed using 3% BSA-PBS (Sigma, St. Louis, MO, USA). TSA visualization was performed with the Opal 520, Opal 620, and Opal 690 (Opal seven-color IHC kit, Perkin Elmer, Waltham, MA, USA). Staining was finished with a DAPI counterstain and slides were enclosed in fluorescence mounting medium (Agilent, Santa-Clara, CA, USA). Slides were scanned using the Vectra 3.0 (PerkinElmer, Waltham, MA, USA). Tissue imaging was performed using inForm Advanced Image Analysis software (inForm 2.1.1 and 2.2.1; Perkin Elmer, Waltham, MA, USA).

### 4.5. RNA-Preserving Immunolabeling and Fluorescence-Guided Laser Microdissection

Seven-micrometer-thick sections of frozen breast tumor samples (*n* = 4) were mounted to PET-frame slides (Carl Zeiss, Oberkochen, Germany) pre-treated by RNAZap (Thermo Fisher Scientific, Waltham, MA, USA). The protocol of immunolabeling has been modified to prevent RNA degradation in the tissue sections. Briefly, the sections were fixed in methanol (Sigma, St. Louis, MO, USA) for 5 min and washed in PBS containing RNAlater (5:1; Thermo Fisher Scientific, Waltham, MA, USA) for 2 min. Thereafter, the sections were incubated for 5 min with the primary antibody cocktail: anti-cytokeratin 7 (CK7, OV-TL, 1:50, Agilent, Santa-Clara, CA, USA) and either anti-KIF14 (HPA038061, 1:500, Sigma, St. Louis, MO, USA) or anti-SPATA18/Mieap (HPA036854, 1:100, Sigma, St. Louis, MO, USA) or EZR (HPA021616, 1:500, Sigma, St. Louis, MO, USA) in PBS:RNAlater (5:1) and washed two times in PBS:RNAlater (5:1) for 2 min. Next, the sections were incubated for 5 min with the secondary antibodies: AlexaFluor 488-conjugated anti-mouse IgG (H+L) and AlexaFluor 555-conjugated anti-rabbit IgG (H+L) diluted in PBS:RNAlater (5:1). Finally, DAPI in PBS:RNAlater (5:1) was used to detect nuclei. The sections were washed two times for 5 s with PBS at room temperature, dehydrated, and air-dried. As control of non-specific binding, sections were incubated with an appropriate primary antibody or with only the secondary antibody. The RNA integrity number (RIN) of the samples obtained from the immunolabeled sections is provided in Appendix A.

Laser capture microdissection guided under fluorescence (PALM, Carl Zeiss, Oberkochen, Germany) was used to isolate CK7-positive KIF14+, KIF14−, Mieap+, Mieap−, EZR+, and EZR− cells from the torpedo-like structures of breast tumors (Appendix A). Approximately 50 samples of each type of cells were isolated from each breast tumor. In total, 24 microdissected specimens were collected.

### 4.6. RNA Extraction, Library Preparation, Sequencing, and Bioinformatic Analysis

RNA was extracted from the microdissected samples using the Single Cell RNA Purification Kit (Norgen, Thorold, ON, Canada). RNA samples were immediately used to generate cDNA libraries using SMARTer Stranded Total RNA-Seq Kit v2—Pico Input Mammalian (Takara, Mountain View, CA, USA). The concentration of cDNA libraries was measured by the dsDNA High Sensitivity kit on a Qubit 4.0 fluorometer (Thermo Fisher Scientific, Waltham, MA, USA) and varied from 1.33 to 15.7 ng/uL. The quality of cDNA libraries was assessed using High Sensitivity D1000 ScreenTape on a 4150 TapeStation (Agilent, Santa-Clara, CA, USA). Libraries were sequenced on a NextSeq500 instrument (Illumina, San Diego, CA, USA) using single-end 75 bp reads.

Raw reads were aligned to the GRCh38 reference genome using STAR v. 2.7.3a (https://github.com/alexdobin/STAR) [58]. The number of mapped reads was counted by the featureCounts tool [59]. The DESeq2 package [60] was used to detect DEGs between KIF14-, Mieap-, and EZR-positive and negative tumor cells. Genes with *p*-values < 0.05 were used for functional enrichment analysis by Enrichr [61,62].

### 4.7. Statistical Analysis

Statistical analysis was performed using Statistica 8.0 (StatSoft, Tulsa, OK, USA) and XLStat (Addinsoft, New York, NY, USA). Pearson’s Chi-square and Fisher’s exact tests were used to analyze the association between expression of HAX1, KIF14, Mieap, and EZR proteins in breast tumors and frequency of distant metastases. The Kaplan–Meier estimator with the log-rank test was used to analyze metastasis-free survival rates of breast cancer patients with expression of HAX1, KIF14, Mieap, and EZR proteins in tumors. Cox proportional hazard analysis was used to assess the association between HAX1, KIF14, Mieap, and EZR expression and metastasis-free survival. Associations were reported as hazard ratios (HRs) with 95% confidence intervals (95% CIs) and *p*-values (likelihood ratio test). Differences were considered significant at *p* < 0.05. Association of DEGs in KIF14-, Mieap- positive and EZR-negative cells with RFS in breast cancer patients was analyzed using Kaplan–Meier plotter [18].

## 5. Conclusions

Our results indicate that the positive expression of KIF14 and Mieap and negative expression of EZR at the tips of the torpedo-like structures is associated with a high frequency of breast cancer metastasis. The underlying mechanism most likely consists of the joint involvement of KIF14- and Mieap-positive cells as well as EZR-negative cells in the realization of breast cancer invasion. In other words, KIF14, Mieap, and EZR can be considered as markers of breast cancer invasion. Altogether, these results again indicate the important role of intratumor morphological heterogeneity in breast cancer prognosis and its potential attractivity as a useful model for the identification of prognostic markers.

## Figures and Tables

**Figure 1 cancers-12-01909-f001:**
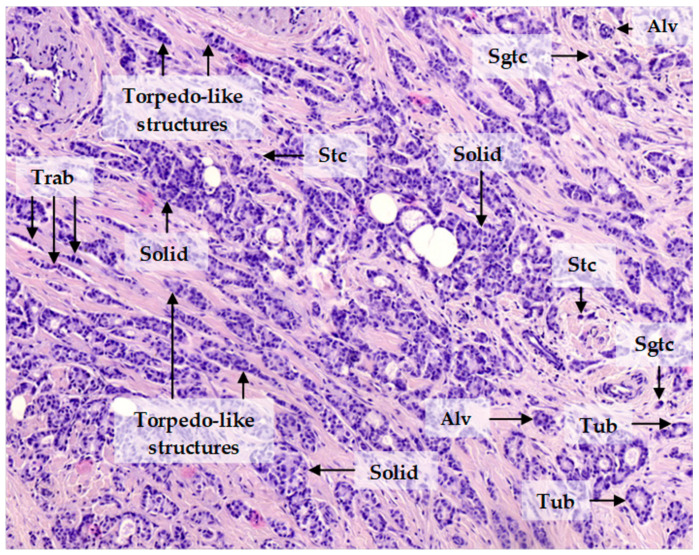
Invasive breast carcinoma of no special type. The invasive component of the tumor is heterogeneous and represented by tubular (Tub), alveolar (Alv), solid, trabecular (Trab), and torpedo-like structures, as well as discrete groups of tumor cells including small groups of two to five tumor cells (Sgtc) and single tumor cells (Stc). Hematoxylin and eosin staining, 200× magnification.

**Figure 2 cancers-12-01909-f002:**
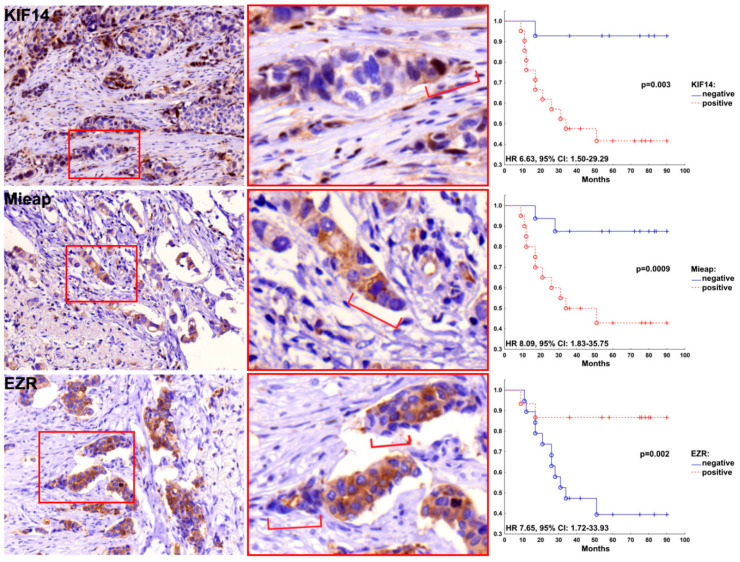
Metastasis-free survival rates of patients with the expression of KIF14, Mieap, and EZR at the tips of the torpedo-like structures in breast tumors. Hematoxylin and Eosin-stained images with 400× magnification.

**Figure 3 cancers-12-01909-f003:**
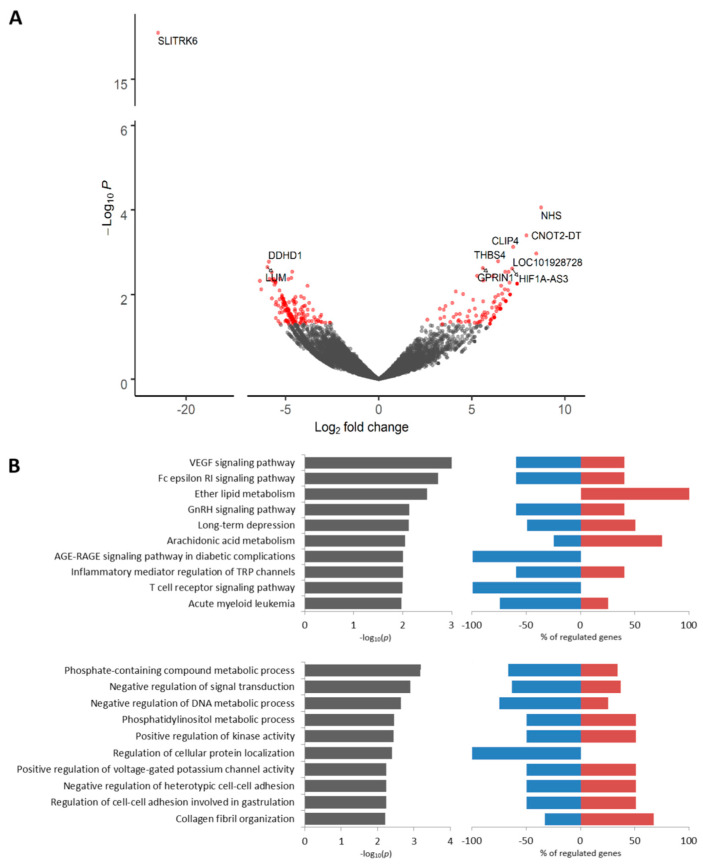
Differential expression analysis of KIF14-positive tumor cells. Volcano plot of TOP 10 up- and downregulated genes (**A**). TOP 10 KEGG signaling pathways and GO biological processes (**B**). Up- and downregulated genes with an unadjusted *p*-value < 0.05 were used for functional enrichment analysis. The left panel shows –log10p of the perturbation determined from a gene set test, whereas the right panel demonstrates a percentage of genes from corresponding KEGG and GO nodes that are down- and upregulated.

**Figure 4 cancers-12-01909-f004:**
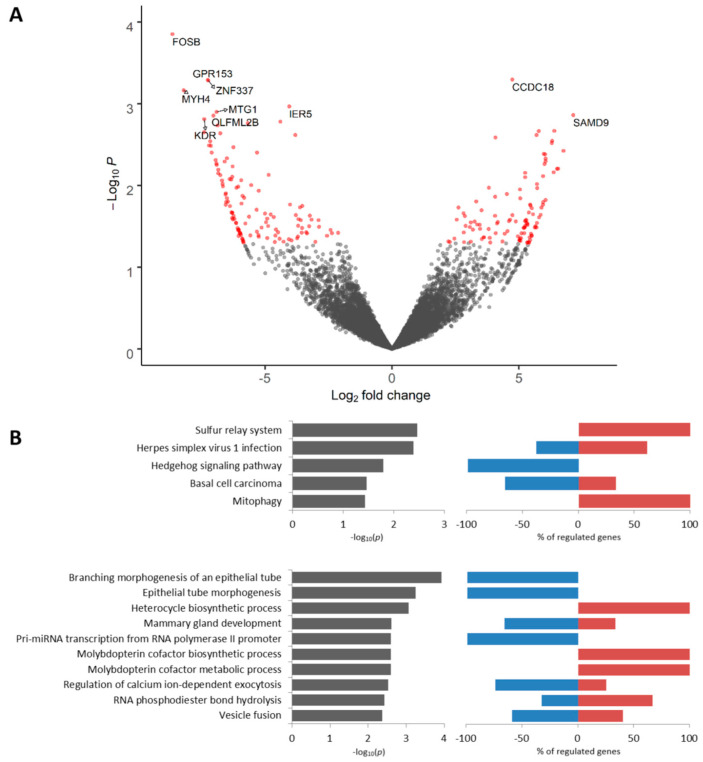
Differential expression analysis of Mieap-positive tumor cells. Volcano plot of TOP 10 up- and downregulated genes (**A**). TOP 10 KEGG signaling pathways and GO biological processes (**B**). Up- and downregulated genes with an unadjusted *p*-value < 0.05 were used for functional enrichment analysis. The left panel shows –log10p of the perturbation determined from a gene set test, whereas the right panel demonstrates a percentage of genes from corresponding KEGG and GO nodes that are down- and upregulated.

**Figure 5 cancers-12-01909-f005:**
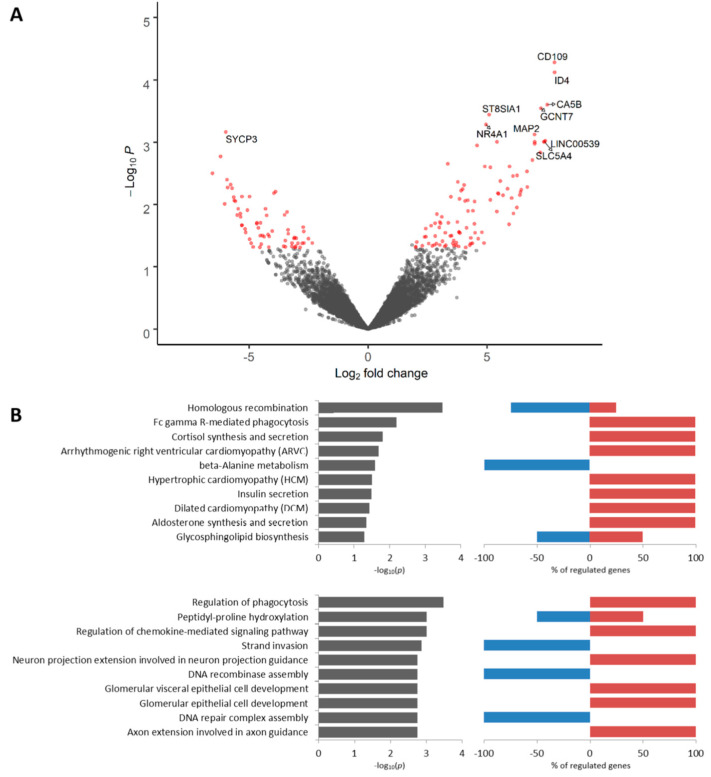
Differential expression analysis of EZR-negative tumor cells. Volcano plot of TOP 10 up- and downregulated genes (**A**). TOP 10 KEGG signaling pathways and GO biological processes (**B**). Up- and downregulated genes with an unadjusted *p*-value < 0.05 were used for functional enrichment analysis. The left panel shows –log10p of the perturbation determined from a gene set test, whereas the right panel demonstrates a percentage of genes from corresponding KEGG and GO nodes that are down- and upregulated.

**Figure 6 cancers-12-01909-f006:**
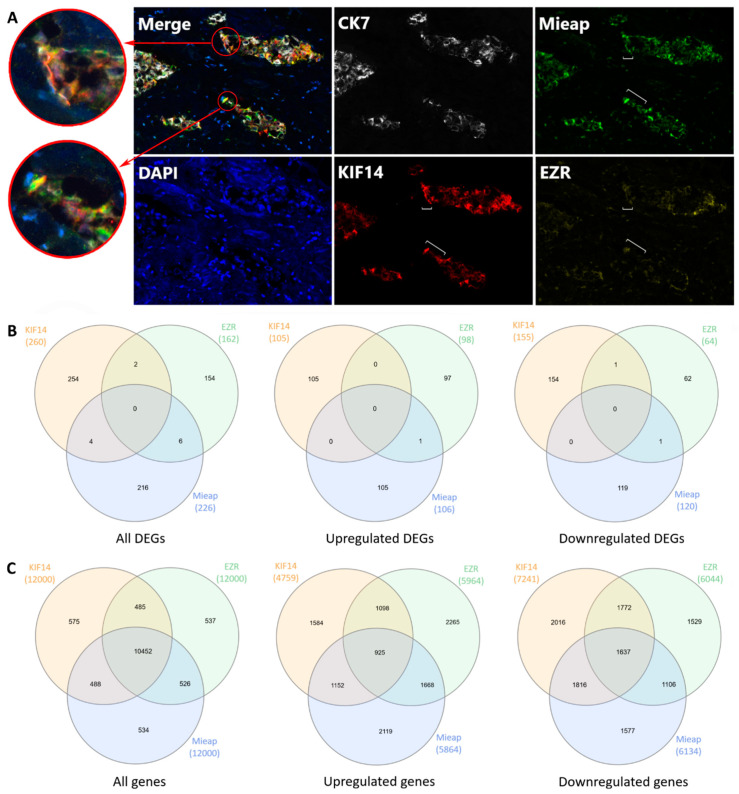
Localization of KIF14-positive, Mieap-positive, and EZR-negative cells in the torpedo-like structures. Expression of KIF14, Mieap, and EZR proteins in the torpedo-like structures. A white bracket indicates the tip of the torpedo-like structures (**A**). Venn diagram summarizing the shared DEGs among KIF14-positive, Mieap-positive, and EZR-negative cells (*p* < 0.05) (**B**). Venn diagram summarizing all genes shared between KIF14-positive, Mieap-positive, and EZR-negative cells (**C**).

**Table 1 cancers-12-01909-t001:** Genes differentially expressed in solid and trabecular structures and associated with cell migration.

Gene	Log-Fold Change					Function	Protein Expression Patterns in Breast Cancer ^‡^
	Tub	Alv	Sol	Trab	Discr		
*HAX1*	0.61	1.14 *	1.42 ^†^	1.27 *	−0.21	Regulator of cortical actin cytoskeleton	Positive expression at the periphery of solid structures
*KIF14*	1.34	2.56	3.87 *	3.42 *	2.73	Microtubule motor protein	Positive expression at the tips of solid structures
*SPATA18* (Mieap)	−1.68 *	−1.62	−2.71 ^†^	−3.02 ^†^	−4.09 ^†^	Regulation of mitochondrial quality and viability	Positive expression at the tips of solid structures
*EZR*	1.73 *	1.09	1.99 ^†^	1.91 ^†^	ND	Connection of major cytoskeletal structures to the plasma membrane	Negative expression at the tips of solid structures

* *p* < 0.1; ^†^
*p* < 0.05; ^‡^ according to the Human Protein Atlas; ND, not determined; Tub, tubular; Alv, alveolar; Sol, solid; Trab, trabecular structures; Discr, discrete groups of tumor cells.

**Table 2 cancers-12-01909-t002:** Frequency of metastases in patients with expression of KIF14, Mieap, and EZR at the tips of torpedo-like structures in breast tumor.

Distant Metastasis	Nuclear Expression of KIF14		Cytoplasmic Expression of Mieap		Cytoplasmic Expression of EZR	
	Positive	Negative	Positive	Negative	Positive	Negative
No	7 (33.3)	12 (85.7)	6 (30.0)	14 (87.5)	13 (86.6)	5 (26.3)
Yes	14 (66.7)	2 (14.3)	14 (70.0)	2 (12.5)	2 (13.3)	14 (73.7)
*p*-value *	0.003		0.001		0.001	

* Fisher’s exact test.

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
