# Peer review of "The Activity of KIF14, Mieap, and EZR in a New Type of the Invasive Component, Torpedo-Like Structures, Predetermines the Metastatic Potential of Breast Cancer"

_cancers, 2020, doi:10.3390/cancers12071909_

Round 1
Reviewer 1 Report
The manuscript “The activity of KIF14, Mieap, and EZR …” by T. Gerashchenko and colleagues, is focused on the establishing a genetic signature that would predetermine the metastatic potential of breast cancer. By correlating the invasive growth patterns of breast tumors with microarray gene expression data, the authors identified two genes, KIF14 and Mieap, the expression of which was positively associated with the presence of invasive, torpedo-like structures in primary tumors. Along with this correlation, the lack of EZR expression in the tips of torpedo-like structures was associated with the high metastatic potential of cancer and poorer outcome for cancer patients. The stratification curves presented in Figure 1 are impressive and support the major conclusion of this study, namely that the established novel association between the levels of KIF14, Mieap, and EZR expression in the tip cells of torpedo-like structures in breast tumors are associated with high metastatic potential of breast cancer an and can be used to predict poorer patient survival.
However, as a general reader of the paper, I would have a problem with the data demonstrating no direct associations between the presence of torpedo-like structures and the breast cancer metastasis and, to some extent, also with the data indicating no correlation between the expression of the three select genes and progression of breast cancer (lines 121-127).
The authors nicely introduced the morphological heterogeneity of breast tumors and explained difficulties in establishing definitive associations between morphological characteristics of primary tumor and development of metastatic disease. In view of the lack of direct links established in the present study between morphological characteristics of primary tumors and their progression towards metastasis (lines 121-124), the authors should explain in a clearer way how their data indicate the importance of recognizing torpedo-like structures for breast cancer patients. The introductory H&E and/or IHC images would help a general reader to appreciate the morphological variety of breast tumors as it relates to the presence or absence of torpedo-like structures in breast cancer tissue. In the present form, Figure 1 just illustrates the expression or the lack of expression of the three selected proteins. This figure would greatly benefit from IHC comparison of protein expression in breast tumors presenting different invasive patters. It would also allow the authors to make a clear conclusion about the LOSS of EZR expression versus the LACK of EZR protein detection. The best representative images from the Supplementary Figure S1 could be used in a main figure to clarify the morphological invasion patterns in primary tumors vs. relative expression of select proteins.
The authors should also consider explaining in more detail the criteria for gene selection: the corresponding section could be moved from Material and Methods to the main text (lines 303-311).
Another concern is related to the necessity of more substantiated clarification that several cancer traits together rather than in their singularity could provide a valuable tool for “metastatic stratification” of breast cancer patients. In the present form, the statements that “breast cancer metastasis was not related to the expression...” of any of select proteins (lines 124-125), significantly dampens the putative significance of the study. The authors should discuss previously published data on gene signatures, indicating that metastasis-predicting genes functionally contributing to metastasis, always involve cooperation of several genes. The research of pro-metastasis genes in lung and breast cancer from the Massague and Weinberg groups can provide a substantial level of support for the data of present study. Thus, the Massague group demonstrated that up to 4 individual genes should be taken into consideration while establishing their involvement in predicting lung cancer metastasis. Finally, I would recommend omitting the last paragraph of Discussion or re-write it in a more constructive manner.
Minor points of concern:
- Since no functional experiments were conducted in the study, the authors should more carefully use the terms indicating the “involvement” of genes instead of their “association” or “correlation” with one or another tumor characteristic ( e.g., lines 79, 88, 211). Similarly, the expression “functional phenotype” should not be applied to data obtained with pure protein expression approaches (line 268).
- Similar consideration should be applied to the statement that the authors demonstrated significant increase in ether lipid metabolism (lines 235-236), since no metabolic studies were performed.
- The expression “ extracted genes expressed in…” should be re-considered since the DNA/RNA material, not genes, was extracted (line 92).
- The declarative expressions about the LOSS of EZR should also be re-considered and used more carefully since the expression of the EXR protein was not followed up in time of tumor progression, or related to the stage/grade of tumor (e.g., line 102).
Author Response
The manuscript “The activity of KIF14, Mieap, and EZR …” by T. Gerashchenko and colleagues, is focused on the establishing a genetic signature that would predetermine the metastatic potential of breast cancer. By correlating the invasive growth patterns of breast tumors with microarray gene expression data, the authors identified two genes, KIF14 and Mieap, the expression of which was positively associated with the presence of invasive, torpedo-like structures in primary tumors. Along with this correlation, the lack of EZR expression in the tips of torpedo-like structures was associated with the high metastatic potential of cancer and poorer outcome for cancer patients. The stratification curves presented in Figure 1 are impressive and support the major conclusion of this study, namely that the established novel association between the levels of KIF14, Mieap, and EZR expression in the tip cells of torpedo-like structures in breast tumors are associated with high metastatic potential of breast cancer an and can be used to predict poorer patient survival.
Reply: We thank the Reviewer for the detailed analysis of our manuscript and helpful comments. The complete Reviewer comments are shown below with our point-by-point responses. Changes to the manuscript were made using the "Track Changes" function.
- However, as a general reader of the paper, I would have a problem with the data demonstrating no direct associations between the presence of torpedo-like structures and the breast cancer metastasis and, to some extent, also with the data indicating no correlation between the expression of the three select genes and progression of breast cancer (lines 121-127).
Reply: We thank the Reviewer to highlight this point. The presence of torpedo-like structures was really not enough to be associated with increased distant metastasis in breast cancer patients. The only expression of KIF14 and Mieap and lack of EZR at the tips of these structures are related to distant metastasis. This can indicate that torpedo-like structures are heterogeneous and only those that have an expression of KIF14 and Mieap and lack of EZR can contribute to metastasis. Interestingly, breast cancer metastasis was not associated with the activity of these proteins in other morphological structures: tubular, alveolar, solid, trabecular, and small groups of tumor cells. These results emphasize that the prometastatic role of KIF14, Mieap, and EZR lack is strongly specific to torpedo-like structures. Future genetic and transcriptomic analysis should reveal the molecular make-up of torpedo-like structures and their connection with other morphological structures. Based on your comment, we modified the text (lines 144-147).
- The authors should explain in a clearer way how their data indicate the importance of recognizing torpedo-like structures for breast cancer patients. The introductory H&E and/or IHC images would help a general reader to appreciate the morphological variety of breast tumors as it relates to the presence or absence of torpedo-like structures in breast cancer tissue. In the present form, Figure 1 just illustrates the expression or the lack of expression of the three selected proteins. This figure would greatly benefit from IHC comparison of protein expression in breast tumors presenting different invasive patters. It would also allow the authors to make a clear conclusion about the LOSS of EZR expression versus the LACK of EZR protein detection.
Reply: We added the introductory H&E image (Figure 1) that comprehensively demonstrates the intratumor morphological heterogeneity in invasive breast carcinoma including such arrangements of tumor cells as tubular, alveolar, solid, trabecular structures, small groups of tumor cells, single tumor cells, as well as torpedo-like structures. Also, we modified Figure 2 (Figure 1 in the first version of the manuscript) where expression of KIF14, Mieap, and EZR is presented in other invasive structures. Finally, we changed “loss of EZR” to “negative expression of EZR” because this phrase seems to be more correct.
- The authors should also consider explaining in more detail the criteria for gene selection: the corresponding section could be moved from Material and Methods to the main text (lines 303-311).
Reply: We moved the criteria for gene selection to the Results (lines 96-104)
- Another concern is related to the necessity of more substantiated clarification that several cancer traits together rather than in their singularity could provide a valuable tool for “metastatic stratification” of breast cancer patients. In the present form, the statements that “breast cancer metastasis was not related to the expression...” of any of select proteins (lines 124-125), significantly dampens the putative significance of the study. The authors should discuss previously published data on gene signatures, indicating that metastasis-predicting genes functionally contributing to metastasis, always involve cooperation of several genes. The research of pro-metastasis genes in lung and breast cancer from the Massague and Weinberg groups can provide a substantial level of support for the data of present study. Thus, the Massague group demonstrated that up to 4 individual genes should be taken into consideration while establishing their involvement in predicting lung cancer metastasis.
Reply: As we mentioned above, the fact that distant metastasis is not associated either with just the presence of torpedo-like structures or KIF14, Mieap, and EZR expression in other morphological structures emphasizes the metastatic significance of the proteins in only torpedo-like structures. The reason for this phenomenon is unclear. Most likely, only torpedo-like structures with an expression of KIF14, Mieap, and EZR are able to “generate metastatic seeds”. We thank the Reviewer to suggest researches by the Massaque and Weinberg groups that were not considered in the first version of the manuscript. We analyzed three works by Massaque et al. which identified a set of genes (EREG, COX2, MMP1, and MMP2) that marks and functionally contribute breast cancer metastasis to lung (Minn et al., Nature 2005; Minn et al., PNAS 2007; Gupta et al., Nature 2007). In this regard, we have modified the Discussion (lines 324-328) and added the following sentence: “In this regard, our results emphasize that several cancer markers together rather than in their singularity could provide a valuable tool for the “metastatic stratification” of breast cancer patients as previously reported by Massague and colleagues [53–55]. In other words, metastasis always involves the cooperation of several genes”.
- Finally, I would recommend omitting the last paragraph of Discussion or re-write it in a more constructive manner.
Reply: We re-wrote the last paragraph of Discussion that contains the study limitations and left only those that are directly related to the present work.
Minor points of concern:
- Since no functional experiments were conducted in the study, the authors should more carefully use the terms indicating the “involvement” of genes instead of their “association” or “correlation” with one or another tumor characteristic ( e.g., lines 79, 88, 211). Similarly, the expression “functional phenotype” should not be applied to data obtained with pure protein expression approaches (line 268).
Reply: The requested changes have been made (lines 27, 85, 94, 253; the expression “functional phenotype” was removed (lines 330, 340)).
- Similar consideration should be applied to the statement that the authors demonstrated significant increase in ether lipid metabolism (lines 235-236), since no metabolic studies were performed.
Reply: We modified this statement (line 279)
- The expression “extracted genes expressed in…” should be re-considered since the DNA/RNA material, not genes, was extracted (line 92).
Reply: Since we moved the criteria for the selection of genes from Materials and Methods to Results, this point was removed.
- The declarative expressions about the LOSS of EZR should also be re-considered and used more carefully since the expression of the EXR protein was not followed up in time of tumor progression, or related to the stage/grade of tumor (e.g., line 102).
Reply: Thank you for this comment. We completely agree that the phrase “loss of EZR” is too declarative and should be supported by additional experiments. In the revised manuscript, we have replaced “loss” with “negative expression” since this term is more common and more appropriate for our situation.
Reviewer 2 Report
The authors present compelling evidence that the expression of KIF14 and Mieap, as well as the loss of EZR expression, are markers for breast cancer metastasis when they appear at the ends of torpedo like structures. They show this using bioinformatics of existing information as well as their own immunohistochemistry and RNA-seq studies. The authors rightly caution that their evidence is compelling but not definitive and suggest appropriate future in vitro experiments to determine the way these genes (or gene loss) contribute to metastasis.
Strenghts and weaknesses:
The introduction frames the state of the field and the aims of the authors well. It seems that there would be more the 9 references to cite about this topic, though, so it could possibly benefit from more citation.
Figure 1 – survival curves are too small, text is almost unreadable. Lines are faint.
Line 174 – make this sentence an X as compared to Y sentence like for the previous gene (lines 155-156). The way lines 174-175 is worded makes it difficult to tell whether you compared positive to negative or negative to positive to get your data. This is doubly important as this is the one with the opposite trend (loss of expression is bad – for the other two, expression is bad). Only when seeing the title of Table S14 did I finally understand that EZR negative was your test case, with EZR positive as control – which is the opposite of the other two analyses.
Lines 192-198 need more explanation. It seems utterly unfeasible that not a single RNA transcript is the same between the three cell populations. Is 162-260 transcripts a decent number for RNA seq from a single cell? It seems unclear how many single cells were sequenced in the same category to obtain adequate transcript coverage. Detecting that few transcripts seems low. It does seem feasible that they didn’t have shared CHANGED genes. But not having any of the same genes overall seems suspect. Also, most of the patients they studied with distant metastasis have 2 or all three markers. Are they in separate cells? Or are there cells positive for both KIF14 and Mieap as well as without EZR? Somewhere in the paper, I feel the need to hear from the author whether they found these mutations in the same cell. In discussion, the authors note three functionally distinct classes of cells at the ends of the torpedo structures. What about the cells that express two of those genes or all three (if those cells exist)?
The authors cite themselves but this is clearly a continuation of previous work, so it doesn’t seem gratuitous.
I very much approve of the cautious way they end the paper. This evidence is not yet enough to change clinical practice. The evidence needs to be more robust, ideally with the mechanism of how these markers affect metastasis.
Author Response
The authors present compelling evidence that the expression of KIF14 and Mieap, as well as the loss of EZR expression, are markers for breast cancer metastasis when they appear at the ends of torpedo like structures. They show this using bioinformatics of existing information as well as their own immunohistochemistry and RNA-seq studies. The authors rightly caution that their evidence is compelling but not definitive and suggest appropriate future in vitro experiments to determine the way these genes (or gene loss) contribute to metastasis.
Reply: We thank the Reviewer for the work on our manuscript and valuable comments. The complete Reviewer comments are shown below with our point-by-point responses. Changes to the manuscript were made using the "Track Changes" function.
Strenghts and weaknesses:
- The introduction frames the state of the field and the aims of the authors well. It seems that there would be more the 9 references to cite about this topic, though, so it could possibly benefit from more citation.
Reply: We added a few new citations in the Introduction (Brabletz et al., Nat. Rev. Cancer 2018;
Nieto et al., Cell 2016; Zavjalova et al., Sib. J. Oncol. 2006)
- Figure 1 – survival curves are too small, text is almost unreadable. Lines are faint.
Reply: According to your comment and suggestions of other Reviewers, we have modified Figure 1 (Figure 2 in the revised manuscript).
- Line 174 – make this sentence an X as compared to Y sentence like for the previous gene (lines 155-156). The way lines 174-175 is worded makes it difficult to tell whether you compared positive to negative or negative to positive to get your data. This is doubly important as this is the one with the opposite trend (loss of expression is bad – for the other two, expression is bad).
Reply: In the case of EZR, we compared negative cells relatively to positive cells, because IHC analysis showed that the negative expression of EZR in the tips of torpedo-like structures is associated with breast cancer metastasis. In the case of KIF14 and Mieap, the situation was the opposite.
- Lines 192-198 need more explanation. It seems utterly unfeasible that not a single RNA transcript is the same between the three cell populations.
Reply: We thank the Reviewer to highlight this important point. In the first version of the manuscript, Venn diagrams showed only DEGs (p<0.05) in KIF14-positive, Mieap-positive, and EZR-negative tumor cells. Up- or downregulated DEGs shared between all three cell populations were not found. But, when we compared all genes expressed in these cells, common genes were detected; however, the overlapping was low (~ 16-25%). These results show again that KIF14-positive, Mieap-positive, and EZR-negative cells are transcriptionally-distinct populations. We added Venn diagrams with all genes in Figure 6.
- Is 162-260 transcripts a decent number for RNA seq from a single cell? It seems unclear how many single cells were sequenced in the same category to obtain adequate transcript coverage.
Reply: The total number of transcripts was ~ 60000 in KIF14-positive, Mieap-positive, and EZR-negative cells. After bioinformatic processing, particularly so-called hard clipping, we restricted to 12000 transcripts. However, DEGs (p<0.05) were much fewer, 160 – 260 depending on the cell type. Using laser microdissection, we isolated ~ 50 samples of each type of cells (positive/negative by studied markers) from each breast tumor. Finally, we got 250 to 500 cells of each type in tubes.
- Detecting that few transcripts seems low. It does seem feasible that they didn’t have shared CHANGED genes. But not having any of the same genes overall seems suspect.
Reply: Our statement was mistaken because we also needed to include Venn-analysis of all genes expressed in three types of cells. In the revised manuscript, we show that despite the absence of common DEGs, 925 upregulated and 1637 downregulated genes are shared between KIF14-positive, Mieap-positive, and EZR-negative cells (Figure 6).
- Also, most of the patients they studied with distant metastasis have 2 or all three markers. Are they in separate cells? Or are there cells positive for both KIF14 and Mieap as well as without EZR? Somewhere in the paper, I feel the need to hear from the author whether they found these mutations in the same cell. In discussion, the authors note three functionally distinct classes of cells at the ends of the torpedo structures.
Reply: The expression of each protein was assessed independently from two other proteins based on the analysis of immunostained sections. In general, each breast tumor sample was used to prepare three sections each of those was stained with antibodies against any one protein. Each immunostained section was analyzed independently from others. After IHC analysis, the expression of each protein was correlated to metastasis frequency and metastasis-free survival. Supplementary Table S16 shows how many patients with metastases have 2 or all three markers in torpedo-like structures. Simultaneous expression of all three proteins was observed in 57.9% (11/19) of the patients with metastases. The co-expression of any two proteins was detected in 15.8% (3/19) of the cases (Table S16). Taking into account that the expression of three proteins was assessed not in the same sections, it is difficult to state whether they expressed in separate/same cells. To answer this question, we performed multiplex IHC and RNA-seq of positive and negative cells. Multiplex IHC showed that these proteins can be both in the same and separate cells. RNA-seq showed the absence of common DEGs and low overlapping if all genes were taken into analysis that may indicate the presence of three transcriptionally-distinct populations of tumor cells.
- What about the cells that express two of those genes or all three (if those cells exist)?
Reply: We think that the Reviewer meant markers (not genes). As mentioned above, all three proteins were observed in 57.9% (11/19) of the patients with metastases. The co-expression of any two proteins was detected in 15.8% (3/19) of the cases (Table S16). If the Reviewer wanted to get information about genes, we would like to know what genes were meant.
- The authors cite themselves but this is clearly a continuation of previous work, so it doesn’t seem gratuitous.
Reply: The Reviewer is right; this manuscript is a continuation of previous work.
- I very much approve of the cautious way they end the paper. This evidence is not yet enough to change clinical practice. The evidence needs to be more robust, ideally with the mechanism of how these markers affect metastasis.
Reply: In the last sentence of the Conclusion, we tried to pay attention that the intratumor morphological heterogeneity could be used as a useful model in future studies focusing on the search for prognostic markers of breast cancer. We modified this sentence.
Reviewer 3 Report
The authors in this study address the intratumoral morphological heterogeneity in breast cancer by mainly focusing on solid and trabecular arrangements. Using gene set enrichment analysis the authors identified few genes that are expressed in solid and trabecular structures and are correlated with disease metastasis and tumor invasion. The study is well designed, however there are certain shortcomings that needs to be addressed before further consideration.
1) How the mutational landscape of particular tumor type alters the expression of these proteins within the tumor bed? Is there any correlation with the primary tumor as well? The authors are requested to provide evidence in this aspect.
2) It is unclear that what accounts for difference in transcriptomic features even though Kif14 and Mieap are expressed in same tumor? The authors are required to provide an explanation and incorporate in discussion section.
3) What accounts for torpedo like phenotype in breast cancer? Is there any specific marker that defines these structures? Is this same sort of adaption mechanism that tumor relies on? The authors are requested to include this in the discussion section
4) The authors have identified several differentially expressed genes in KIF14 and Mieap positive tumor cells. However, none of these targets are validated in this study. The authors should provide some sort of experimental evidence suggesting that these genes are correlated with tumor metastasis in either solid or torpedo structures.
5) In vitro data is missing in this study. It will be interesting to see how the expression levels of these proteins correlate with EMT gene signatures? The authors are requested to provide experimental evidence in support of this.
Author Response
The authors in this study address the intratumoral morphological heterogeneity in breast cancer by mainly focusing on solid and trabecular arrangements. Using gene set enrichment analysis the authors identified few genes that are expressed in solid and trabecular structures and are correlated with disease metastasis and tumor invasion. The study is well designed, however there are certain shortcomings that needs to be addressed before further consideration.
Reply: We thank the Reviewer for the careful reading of our manuscript and important comments. The complete Reviewer comments are shown below with our point-by-point responses. Changes to the manuscript were made using the "Track Changes" function.
- How the mutational landscape of particular tumor type alters the expression of these proteins within the tumor bed? Is there any correlation with the primary tumor as well? The authors are requested to provide evidence in this aspect.
Reply: In this study, we did not analyze the mutational profile of KIF14-positive, Mieap-positive, and EZR-negative tumor cells. The mechanisms underlying the altered expression of these proteins at the tips of torpedo-like structures are unknown. We can assume that genetic alterations do not affect the expression of these genes since they are very rarely mutated in breast cancer according to TCGA. We added this information in the manuscript (lines 315-318).
- It is unclear that what accounts for difference in transcriptomic features even though Kif14 and Mieap are expressed in same tumor? The authors are required to provide an explanation and incorporate in discussion section.
Reply: We thank the Reviewer for this very important question. Now, it is difficult to say why transcriptomic features differ between KIF14-positive, Mieap-positive, and EZR-negative tumor cells. Most likely, non-genetic factors including epigenetic modifications, transcriptional, translational, and post-translational regulation can be involved in the regulation of expression of KIF14, Mieap, and EZR in torpedo-like structures. This regulation can be promoted, for example, by tumor microenvironmental signals. In addition, these cells can acquire their functional features during specialization in collective cell invasion as previously shown by others (Zhang et al., PNAS 2019; Aoki et al., Cancer Sci 2019, etc.). However, further research is needed to understand these mechanisms. We added the discussion of potential mechanisms of the regulation of KIF14, Mieap, and EZR expression in the manuscript (lines 315-318).
- What accounts for torpedo like phenotype in breast cancer? Is there any specific marker that defines these structures? Is this same sort of adaption mechanism that tumor relies on? The authors are requested to include this in the discussion section
Reply: It is an important question that is still unanswered. We think that torpedo-like structures are one of the manifestations of breast cancer invasive growth. There is no specific molecular marker that defines torpedo-like structures. However, they can be identified by routine morphological (hematoxylin and eosin) analysis of breast tumor samples. The origin of these structures, their evolution, and their place in tumor adaptation are subjects for further research.
- The authors have identified several differentially expressed genes in KIF14 and Mieap positive tumor cells. However, none of these targets are validated in this study. The authors should provide some sort of experimental evidence suggesting that these genes are correlated with tumor metastasis in either solid or torpedo structures.
Reply: In this study, we used RNA-seq to show how much KIF14-positive, Mieap-positive, and EZR-negative cells located at the tips of torpedo-like structures differ from each other. Of course, during this analysis, we identified interesting DEGs that can predetermine the phenotype of the studied cells and their metastatic potential. However, it was beyond the scope of the present study. Nevertheless, using the Kaplan-Meyer Plotter database, we assessed whether the expression of the identified DEGs is associated with recurrence-free survival in breast cancer patients. The corresponding results have been added in the manuscript (lines 162-164, 184-186, 204-206, 285-287, 297-299, 313-314), plots – in the Supplementary (Figure S5-S7).
- In vitro data is missing in this study. It will be interesting to see how the expression levels of these proteins correlate with EMT gene signatures? The authors are requested to provide experimental evidence in support of this.
Reply: Undoubtedly, an in vitro study is required to confirm the obtained results. However, previous in vitro studies have already shown the involvement of KIF14, Mieap, and EZR in cell migration and invasion. These works are highlighted in the Discussion. However, it would be more useful to develop a 3D model that mimics the intratumor morphological heterogeneity of breast cancer and to show the effect of these proteins on collective invasion, particularly in torpedo-like structures. In regard to EMT, we tried to assess the epithelial-mesenchymal degree of the studied cells using two algorithms developed by our collaborator, Dr. Mohit Kumar Jolly. The differences of cells in the EMT level were not significant, and we decided not to include these results in the manuscript. Briefly, a tendency toward a more mesenchymal phenotype was observed in EZR-negative cells, whereas the hybrid epithelial-mesenchymal phenotype was specific to KIF14-positive cells. If the Reviewer considers that these results should be included in the manuscript, we will do it.
Reviewer 4 Report
In the current article, the authors used the intrinsic heterogeneity of Breast Cancers to identify molecules involved in invasion and metastasis. Among human BC samples, they have identified genes differentially expressed in solid, trabecular, and in other different structures of tumor cells by gene expression microarray. They performed Immunohistochemistry assay to validate the association of the selected genes and the related proteins expressed. RNA-sequencing was performed to stratify metastatic tumor cells in the torpedo-like structures.
The analysis identified three interesting co-regulated genes and proteins with a dynamic role in metastasis: KIF14 (microtubule motor protein) and Mieap (regulator of mitochondrial viability), upregulated in the tips of the trabecular structures, and EZR (connector of cytoskeleton elements to plasma membrane) downmodulated within the same structures.
KIF14, Mieap, and EZR were found to be expressed within the same or in different cells of the trabecular structures, suggesting cooperation between invading cancer cells.
The experiments described were conducted with solid approaches and exhaustive demonstrations. The paper is clear, well written, well documented, and comes with novel conclusions, supported by the analysis of the data presented; therefore, it deserves publication.
Just few suggestions to better deepen some intriguing points.
- Firstly, it might be of interest to characterize the co-expression of widely accepted markers for invasive breast carcinoma in all KIF14+, Mieap+, EZR- cells on the tips of torpedo-like structures (by Immunoistochemistry or, even more helpful, colocalization by Immunofluorescence) in order to enrich and emphasize the concept.
- Ki67 is considered to be a protein associated with cell cycle and shows significant correlation with the growth of tumor masses and has been proposed as a prognostic or predictive marker in human BC. Ki67 over-expression distribution at the invasive front of BC is associated with bone and liver metastasis. The authors could analyze whether tumor tissues present overexpression of nuclear Ki67 proteins in the tip cells of the torpedo-like structures selected.
- The authors may also analyze the MMPs expression profile: MMPs such as MMP-2, MMP-9, MMP-13, are known to be involved in the invasion and metastasis of breast cancer and are typically overexpressed on the invasive front of carcinomas to degrade extracellular matrix. An experiment should be planned to capture by IHC/IF the expression level and localization of the MMPs. The presence of co-expression by immunofluorescence with the selected proteins should be verified to test the pro-invasive potential of cancer cells in the torpedo-like structures mentioned.
- Another system that induces the degradation of the ECM components is the plasminogen activation system through the serine protease plasmin. The system is composed by urokinase-type plasminogen activator (uPA) and its receptor (uPAR), the interaction of which stimulates proteolysis of plasminogen to plasmin. uPA and uPAR are prognostic marker in breast cancer, therefore the authors may also verify the expression of the uPAS components (uPA, uPAR, PA-1) on the tip of trabecular structures, in the KIF14+, Mieap+, EZR- cells.
It would be helpfull either to test at least one of the mentioned markers unless (maybe the authors have already ran such experiments?) or to discuss this point.
- An intriguing emerging data is represented by the gene expression regulation of those cells co-overexpressing KIF14-Mieap and simultaneously dowmodulating EZR (shown in Figure 5). The RNA-sequencing of cells singly overexpressing either KIF14 or Mieap and cells singly downregulating EZR showed no overlapping genes (Figure 5B). On the other hand, the authors stated that KIF14 and Mieap expression and EZR loss was co-localized in a subset of cells. Nevertheless, in the analysis presented in fig 5B is not clear the contribution of this population (KIF14-Mieap positive and simultaneously EZR negative) since three populations of cells have been always analyzed as separate pools of cells (KIF14 alone, Mieap alone, and EZR alone). Considering this heterogeneous pattern, please explain better and discuss this issue and how is possible not to find matches in gene regulated between the three groups of cells since they could for some extent overlap.
Minor points:
Figure 1 :
- Enlarge the text size below the survival plots.
- The loss of expression of EZR at the tips of the torpedo-like structures is not as evident as the KIF14 and Mieap overexpression. In the selected image I hardly can recognize EZR- cells in one of the two trabecular structures (choose another image or provide a magnification within the panel).
Figure 5:
- Provide a magnification of the co-expressing cells in the merged image and mark the outlines of the trabecular structures to be distinguished from the surrounding tissue.
Author Response
In the current article, the authors used the intrinsic heterogeneity of Breast Cancers to identify molecules involved in invasion and metastasis. Among human BC samples, they have identified genes differentially expressed in solid, trabecular, and in other different structures of tumor cells by gene expression microarray. They performed Immunohistochemistry assay to validate the association of the selected genes and the related proteins expressed. RNA-sequencing was performed to stratify metastatic tumor cells in the torpedo-like structures.
The analysis identified three interesting co-regulated genes and proteins with a dynamic role in metastasis: KIF14 (microtubule motor protein) and Mieap (regulator of mitochondrial viability), upregulated in the tips of the trabecular structures, and EZR (connector of cytoskeleton elements to plasma membrane) downmodulated within the same structures.
KIF14, Mieap, and EZR were found to be expressed within the same or in different cells of the trabecular structures, suggesting cooperation between invading cancer cells.
The experiments described were conducted with solid approaches and exhaustive demonstrations. The paper is clear, well written, well documented, and comes with novel conclusions, supported by the analysis of the data presented; therefore, it deserves publication.
Reply: We thank the Reviewer for positive comments on our manuscript. The complete Reviewer comments are shown below with our point-by-point responses. Changes to the manuscript were made using the "Track Changes" function.
Just few suggestions to better deepen some intriguing points.
- Firstly, it might be of interest to characterize the co-expression of widely accepted markers for invasive breast carcinoma in all KIF14+, Mieap+, EZR- cells on the tips of torpedo-like structures (by Immunoistochemistry or, even more helpful, colocalization by Immunofluorescence) in order to enrich and emphasize the concept.
- Ki67 is considered to be a protein associated with cell cycle and shows significant correlation with the growth of tumor masses and has been proposed as a prognostic or predictive marker in human BC. Ki67 over-expression distribution at the invasive front of BC is associated with bone and liver metastasis. The authors could analyze whether tumor tissues present overexpression of nuclear Ki67 proteins in the tip cells of the torpedo-like structures selected.
- The authors may also analyze the MMPs expression profile: MMPs such as MMP-2, MMP-9, MMP-13, are known to be involved in the invasion and metastasis of breast cancer and are typically overexpressed on the invasive front of carcinomas to degrade extracellular matrix. An experiment should be planned to capture by IHC/IF the expression level and localization of the MMPs. The presence of co-expression by immunofluorescence with the selected proteins should be verified to test the pro-invasive potential of cancer cells in the torpedo-like structures mentioned.
- Another system that induces the degradation of the ECM components is the plasminogen activation system through the serine protease plasmin. The system is composed by urokinase-type plasminogen activator (uPA) and its receptor (uPAR), the interaction of which stimulates proteolysis of plasminogen to plasmin. uPA and uPAR are prognostic marker in breast cancer, therefore the authors may also verify the expression of the uPAS components (uPA, uPAR, PA-1) on the tip of trabecular structures, in the KIF14+, Mieap+, EZR- cells.
It would be helpfull either to test at least one of the mentioned markers unless (maybe the authors have already ran such experiments?) or to discuss this point.
Reply: We thank the Reviewer for this suggestion. At first, we checked if an expression of the MKI67, PLAU, PLAUR, MMP2, MMP9, and MMP13 genes differs between in KIF14-positive, Mieap-positive, EZR-negative cells, on one side, and KIF14-negative, Mieap-negative, and EZR-positive cells, on another side (Table S17). We found the significant downregulation of only the MMP13 gene in KIF14 positive cells (p = 0.03). Afterwards, we used multiplex IHC to demonstrate if MMP13 expression differs between cells composing torpedo-like structures. However, all tumor cells within the torpedo-like structure and in other neighboring structures were MMP13-positive (Figure S8).
- An intriguing emerging data is represented by the gene expression regulation of those cells co-overexpressing KIF14-Mieap and simultaneously downmodulating EZR (shown in Figure 5). The RNA-sequencing of cells singly overexpressing either KIF14 or Mieap and cells singly downregulating EZR showed no overlapping genes (Figure 5B). On the other hand. the authors stated that KIF14 and Mieap expression and EZR loss was co-localized in a subset of cells. Nevertheless. in the analysis presented in fig 5B is not clear the contribution of this population (KIF14-Mieap positive and simultaneously EZR negative) since three populations of cells have been always analyzed as separate pools of cells (KIF14 alone. Mieap alone. and EZR alone). Considering this heterogeneous pattern. please explain better and discuss this issue and how is possible not to find matches in gene regulated between the three groups of cells since they could for some extent overlap.
Reply: The statement about no overlapping genes between KIF14-positive, Mieap-positive, and EZR-negative cells was not correct enough because it was true only for DEGs (p < 0.05). When we took in Venn-analysis all genes up- and downregulated in these cells, common genes were found (925 upregulated and 1637 downregulated). Nevertheless, the percentage of overlapped genes was low (~ 16-25%). Based on this, we modified Figure 6 and subsection “KIF14- and Mieap-positive and EZR-negative cells are co-localized in torpedo-like structures” (lines 225-228). In general, taking into account no overlapping of DEGs and a low number of common genes (in the case of Venn-analysis of all genes), we think that KIF14- and Mieap-positive and EZR-negative cells are transcriptionally-distinct tumor cells. However, we don’t still know if KIF14- and Mieap-positive and EZR-negative cells interact with each other in torpedo-like structures and whether their cooperation is needed for breast cancer invasion and metastasis. In addition, we could say exactly what mechanisms are involved in the regulation of expression of KIF14, Mieap, and EZR in torpedo-like structures. Most likely, genetic alterations are not responsible for changes in their expression because these genes are very rarely mutated in breast cancer. Further studies are needed to get answers to the above questions.
Minor points:
- Figure 1: Enlarge the text size below the survival plots.
Reply: We have enlarged the text size.
- The loss of expression of EZR at the tips of the torpedo-like structures is not as evident as the KIF14 and Mieap overexpression. In the selected image I hardly can recognize EZR- cells in one of the two trabecular structures (choose another image or provide a magnification within the panel).
Reply: We modified Figure 2 (Figure 1 in the first version of the manuscript) by providing the original image with KIF14, Mieap, and EZR staining not only in torpedo-like structures but also in other invasive patterns and magnification of torpedo-like structures with positive expression of KIF14 and Mieap and negative expression of EZR in their tips.
- Figure 5: Provide a magnification of the co-expressing cells in the merged image and mark the outlines of the trabecular structures to be distinguished from the surrounding tissue.
Reply: We have provided a magnification of the co-expressing cells in the merged image.